# Prediction of cognitive impairment through speech data analysis: A comparative evaluation of deep learning models

Minsoo Kim[1], Hyunjoo Choi[2], YongSoo Shim[3], Nayoung Ryoo[3], Ho Tae Jeong[4], Gihyun Yun[1], Hunboc Lee[1], SangYun Kim[5], Young Chul Youn[4,6,7]*

1 Research and Development, Baikal AI Inc., Seoul, Republic of Korea, 2 Department of Communication Disorders, Korea Nazarene University, Cheonan, Republic of Korea, 3 Department of Neurology, The Catholic University of Korea, Seoul, Republic of Korea, 4 Department of Neurology, Chung-Ang University Hospital, Seoul, Republic of Korea, 5 Department of Neurology, Seoul National University College of Medicine and Seoul National University Bundang Hospital, Seongnam-si, Gyeonggi-do, Republic of Korea, 6 Department of Neurology, Chung-Ang University College of Medicine, Seoul, Republic of Korea, 7 Department of Medical Informatics, Chung-Ang University Hospital, Seoul, Republic of Korea

* neudoc@cau.ac.kr

## Abstract

### Background

The early detection of cognitive impairments, such as mild cognitive impairment (MCI) and Alzheimer's disease (AD), is essential for timely intervention and management. This study evaluates the performance of various deep-learning models in classifying speech recordings from individuals with normal cognition (NC), MCI, and AD, to identify the most effective approach for audio-based cognitive impairment diagnosis.

### Methods

Speech data were obtained from the AI Hub "Cognitive Impairment Diagnosis Voice/ Conversation" dataset. The study analyzed voice recordings from 320 female participants (105 with Alzheimer's disease, 92 with mild cognitive impairment, and 123 cognitively normal controls). Three deep-learning architectures were compared: a one-dimensional convolutional neural network (1D CNN), an audio spectrogram transformer (AST), and a speech recognition model (Wav2Vec 2.0). The models were trained using features such as spectrograms, mel-spectrograms, and mel-frequency cepstral coefficients (MFCCs). Model performance was assessed using accuracy, recall, precision, and F1-score, with a five-fold cross-validation strategy to ensure robust and unbiased evaluation. Statistical significance was assessed using pairwise proportion z-tests with Holm-Bonferroni correction, and Wilson score 95% confidence intervals were computed for each model's accuracy.

**Data availability statement:** Data Availability: The original audio dataset used in this study is available to approved researchers through the Korean AI Hub Safe Zone (https://safe-zone.aihub.or.kr/). Due to participant privacy regulations, unrestricted public distribution of the original audio recordings is not possible. However, the data is accessible to researchers upon application for scholarly purposes. To reproduce our findings, interested researchers can formally access the identical dataset by registering at the AI Hub Safe Zone and agreeing to the ethical terms of use. The core source code necessary for reproducing our study's findings has been deposited on Figshare (https://figshare.com/s/3d395292a035369d-4bac). The deposited code includes complete implementations of all 11 model architectures (1D CNN, AST, Wav2Vec 2.0, and 6 ImageNet-pretrained CNNs), the training pipeline, audio preprocessing pipeline, and evaluation utilities.

**Funding:** This work was supported by the Tech Incubator Program for Startup Korea (TIPS) through the Korea Technology and Information Promotion Agency for SMEs (TIPA) funded by the Ministry of SMEs and Startups (MSS, Republic of Korea) (Project No. S3079103) and the Cooperative Research Program for Agriculture Science and Technology Development Rural Development Administration (Project No. PJ01712403). There was no additional external funding received for this study. No, the sponsors/funders had no role in the study design, data collection and analysis, decision to publish, or preparation of the manuscript. The research was conducted independently by the authors.

**Competing interests:** The authors have declared that no competing interests exist.

## Results

Wav2Vec 2.0 outperformed the other models, achieving the highest accuracy and F1-scores for NC vs. MCI (accuracy: 0.74, F1: 0.72) and NC vs. AD (accuracy: 0.83, F1: 0.83). Pairwise proportion z-tests with Holm-Bonferroni correction confirmed that Wav2Vec 2.0 significantly outperformed 7 of 10 competing models (corrected $p < 0.05$) in both classification tasks. Performance varied by model and classification task, with Wav2Vec 2.0 consistently demonstrating superior accuracy across labels.

## Conclusion

This study emphasizes the importance of selecting appropriate models and features for task-specific optimization and provides a foundation for developing non-invasive, speech-based diagnostic tools for cognitive disorders.

## Introduction

The early detection and diagnosis of cognitive impairments, such as mild cognitive impairment (MCI) and Alzheimer's disease (AD), play a pivotal role in managing their progression and treatment [1,2]. MCI represents an intermediate stage between normal aging and dementia, characterized by noticeable cognitive decline without significant interference in daily activities [3]. It is a risk factor for AD, with an estimated 10–15% of MCI cases progressing to AD annually [4]. However, MCI can also remain stable or even revert to normal cognition, making accurate detection and classification challenging [5,6]. AD, on the other hand, is a progressive neurodegenerative disorder and the most common cause of dementia, accounting for 60–70% of cases. It is marked by the accumulation of amyloid-beta plaques and tau tangles in the brain, leading to severe cognitive and functional decline [7]. The socioeconomic burden of AD is immense. Early and accurate diagnosis is crucial for managing disease progression and improving patient outcomes, especially as therapeutic advancements focus on preclinical and early-stage interventions.

Biomarkers, such as those extracted from magnetic resonance imaging, cerebrospinal fluid analysis, and positron emission tomography, can accurately distinguish AD from non-affected individuals [8,9]. However, these diagnostic methods result in high costs for healthcare systems and can impose financial and emotional burdens on patients, limiting their widespread use in screening settings [10]. Consequently, there is a growing need for non-invasive, cost-effective diagnostic tools that can identify cognitive impairments at an early stage.

Recent advancements in artificial intelligence (AI) and machine learning have enabled the use of speech and vocal biomarkers for establishing non-invasive and efficient diagnostic tools [11,12]. Speech data contain valuable information for understanding human cognition and identifying potential cognitive impairments, as changes in language fluency, prosody, and coherence can reflect underlying neural deficits associated with MCI and AD. Thus, AI-driven approaches, particularly those

employing deep-learning techniques, have demonstrated promising results in processing speech data to detect subtle cognitive changes. Various model-training methods, such as deep-learning techniques, have been used to leverage speech characteristics for analysis.

Recent studies have demonstrated the potential of speech data as a non-invasive biomarker for detecting cognitive impairments. Yang et al. (2022) provided a comprehensive review of deep learning methods applied to speech analysis for Alzheimer's detection, highlighting the efficacy of convolutional and recurrent neural networks in capturing temporal features of speech [13]. Similarly, Fristed et al. (2021) explored the use of smartphone-based AI systems for remote early detection of AD, demonstrating the feasibility of real-world applications in non-clinical settings [12].

While many studies have focused on specific architectures, such as convolutional neural networks (CNNs) or recurrent neural networks (RNNs), recent advances in transformer-based models have opened new possibilities. For instance, Gong et al. (2021) introduced the Audio Spectrogram Transformer (AST) for audio classification tasks, showing its ability to leverage self-attention mechanisms for enhanced feature extraction [14]. Baevski et al. (2020) proposed Wav2Vec 2.0, a self-supervised learning framework for speech representation, which has been shown to outperform traditional models in various speech recognition and classification tasks [15]. Despite these advancements, there remains a need for a systematic evaluation of these methodologies in the context of cognitive impairment detection.

This study was aimed at identifying the most suitable approach for analyzing speech data for predicting cognitive impairment. To this end, we performed a comparative evaluation of different model-training methods to assess their effectiveness in predicting cognitive impairment.

In general, several model-training methods have been applied when using voice for AI analysis. The optimal method, however, depends on the application field and objectives; representative examples of such methods include CNNs, RNNs, and transformers [16].

Although CNNs are known for their image processing capabilities, especially in deep-learning contexts, they can also be effectively applied to voice data. Unlike two-dimensional (2D) CNNs used in image processing, which handle three-dimensional data comprising width, height, and channels, this study employed a one-dimensional (1D) that can effectively process time-series data such as audio. Specifically, a 1D CNN can capture unique patterns within a spectrogram's frequency range that may indicate cognitive impairments [17]. This model slides a 1D kernel along the time axis of a spectrogram, thereby learning to capture and extract specific patterns and features related to cognitive impairments within various frequency ranges. Based on the extracted speech patterns and features, it is possible to differentiate cognitive impairments. Given that a spectrogram encompasses the majority of voice features, including pause, pitch, and formant, we anticipated that cognitive impairments could be classified from the variations or outliers in speech patterns. This methodological transition from 1D sequential audio signals to structured 2D visual representations (spectrograms) for complex pattern recognition closely mirrors advanced signal processing paradigms emerging in other biomedical domains, such as utilizing 1D-to-2D transformations coupled with CNNs for radio frequency-based emotion identification [18].

The CNN-architecture-based deep-learning models considered in this study included AlexNet, Inception, visual geometry group-19 (VGG-19), residual network (ResNet), DenseNet, and SqueezeNet [19,20]. AlexNet, introduced in 2012, revolutionized deep learning by overcoming the limitations of previous CNN models. Inception (or GoogLeNet) was designed with a complex architecture to improve the computational efficiency of CNNs. VGG-19, a model comprising 19 layers, demonstrated the performance enhancement achievable using deep networks. ResNet incorporated residual blocks to address learning problems encountered in deep networks. DenseNet used dense connections to facilitate efficient learning by directly connecting features in the depth direction. SqueezeNet was designed to reduce the number of parameters while maintaining a performance comparable with that of AlexNet, thereby increasing the computational efficiency.

Transformers, a recent innovation, have also gained significant attention in the field of voice analysis [21]. Transformers use self-attention mechanisms to process sequential data, thus proving valuable in tasks such as voice recognition, machine translation, and speech synthesis. We considered the audio spectrogram transformer (AST) model [14], a

variant of the vision transformer (ViT) model specialized for audio data [22]. The AST demonstrated that a model trained on extensive image data can also be applied to tasks involving spectrograms, which are visual representations of audio inputs. Originally designed for image classification tasks, the ViT model learns from a vast array of image data. The AST model leverages this capability by converting audio inputs into spectrogram image formats and applying them to audio classification tasks.

Wav2Vec 2.0, a speech recognition model, leverages an autoencoder framework to encode speech signals using a CNN and then processes these signals through a transformer architecture to generate speech embeddings [15,23]. Wav-2Vec 2.0 employs self-supervised learning, allowing it to learn from unlabeled data and be pretrained on a vast corpus of speech samples from various languages. In this manner, Wav2Vec 2.0 can effectively capture the nuances of speech, resulting in enhanced performance in downstream tasks, even with limited data. Its capability to extract critical features from speech without manual feature engineering marks a substantial advancement over traditional methods, rendering it a robust solution for diverse speech-processing applications.

In this study, we aimed to explore the potential of various deep-learning architectures, specifically 1D CNN, AST, and Wav2Vec 2.0, for predicting cognitive impairment. These models were used to classify cognitive states associated with MCI, AD, and normal cognition (NC), based on detailed speech features such as mel-spectrograms. The objective was to enhance the ability of these models to accurately distinguish between these conditions. Despite the inherent differences in architecture and pretraining processes across the models, each model was meticulously optimized and applied in the study context to ensure the most effective use of its structural and functional capacities. By comparing various model-training methods for analyzing speech data, this study makes a significant contribution to the existing body of knowledge by identifying the most effective approach for predicting cognitive impairments, providing valuable insights for future research and mobile applications.

## Materials and methods

### Dataset

The dataset used in this study was acquired from the "Cognitive Impairment Diagnosis Voice/Conversation" offline data on 07/09/2022, which can be accessed at the AI Hub Safe Zone (https://safezone.aihub.or.kr/). This dataset contains voice data from individuals with dementia and control groups, collected through the SpeechTaskSet application. This clinical app is structured to guide participants through 11 heterogeneous speech tasks, including sentence repetition (three tasks), image description (two tasks), language fluency (two tasks), calculation (one task), and storytelling (three tasks), producing an audio file of up to 1 min per task. Notably, the app captures only the voices of the participants, excluding any input from the instructors. To generate sufficient temporal data for deep learning algorithms, the audio recordings spanning all 11 speech tasks were aggregated and utilized collectively per participant.

The participants included in this study were required to meet the following inclusion criteria: (1) a confirmed diagnosis of AD, MCI, or NC based on clinical assessments, (2) aged 50 years or older, and (3) native speakers of the Korean language used in the dataset. The exclusion criteria were: (1) presence of other neurological disorders affecting speech (e.g., Parkinson's disease, stroke), (2) severe hearing impairment, and (3) significant speech disorders unrelated to cognitive impairment.

This study was conducted in accordance with ethical standards, following a protocol reviewed and approved by the Institutional Review Board (IRB). The protocol was registered with the public IRB (https://public.irb.or.kr/) under management number 2022-0642-001 and received approval number P01-202207-01-001. The IRB approval process ensured that all research activities complied with national regulations and guidelines for the ethical treatment of participants.

The AI Hub dataset included 3,483 voice recordings from 320 female participants and 1,481 from 136 male participants. Owing to the lower volume of male recordings, this study prioritized the female voice dataset for model training to prevent extreme class imbalance. A total of 320 female participants were analyzed, including 105 with AD, 92 with MCI,

and 123 cognitively normal controls (NC) (Table 1). The recordings from the AD group corresponded to an average age of 72 years (57–90 y). The corresponding value for the MCI group was 75 years (53–93 y), and for the NC group, it was 69 years (54–88 y).

Audio was sampled at 16 kHz, which is more than two times the threshold of 6800 Hz. According to the Nyquist–Shannon sampling theorem, this value was set to ensure that various elements and characteristics of speech, potentially pivotal for accurately classifying cognitive impairments, could be captured.

## Audio preprocessing

The audio preprocessing pipeline consisted of the following sequential steps applied to all recordings prior to feature extraction: (1) Audio loading and resampling: All audio files were loaded as 32-bit floating-point signals using the soundfile library and resampled to 16 kHz using librosa. Multi-channel recordings were converted to mono by channel averaging. (2) Bandpass filtering: A fifth-order Bessel bandpass filter (300–4000 Hz) was applied to all audio signals to attenuate environmental noise and frequency components outside the human speech frequency range. This filtering served as the primary signal cleaning mechanism, constraining the frequency band to the speech-relevant range. (3) Duration filtering: Recordings shorter than 3 seconds were excluded based on the empirical observation that such brief recordings are disproportionately affected by background noise relative to speech content. (4) Feature-level normalization: During spectrogram generation, the magnitude spectrum was converted to a power spectrum and subsequently transformed to the decibel (dB) scale. Reference normalization against the maximum power was applied (amin = $1 \times 10^{-16}$), the dynamic range was clipped to 80 dB, and the resulting values were linearly normalized to the [0, 1] range.

No explicit silence trimming was applied, as pause patterns during speech—including their duration, frequency, and distribution—were considered potentially informative features of cognitive impairment and were therefore preserved in the model input.

## Features

We used acoustic features including spectrograms, mel spectrograms, and mel-frequency cepstral coefficients as inputs to the models for cognitive disorder classification.

The spectrogram, visually representing the spectrum of frequencies in a sound signal as they change over time [24], was generated using the short-time Fourier transform [25]. Specifically, the audio signal was broken into smaller, overlapping windows that were then subjected to the Fourier transform. We set the window length as 0.025 s and hop length as 0.01 s. The Hann function was used as the window function [26], and the non-equispaced fast Fourier transform (NFFT) [27] was set as 512, aiming to construct a spectrogram best fitted for cognitive disorder classification from voice data. Frequency bands above 4000 Hz, considered unrelated to the voice, were discarded from the spectrogram.

To account for the nonlinear frequency sensitivity of the human auditory system, we used mel spectrograms [28], which transform the frequency scales of spectrograms into the mel scale. In general, the mel scale effectively models the critical perception of different frequencies by the human ear, approximately linear at lower frequencies and increasingly compressed at higher frequencies. This feature provided a more perceptually relevant representation of the audio signal.

Table 1. Characteristics of the voice dataset.

| Group | Number of Datapoints | Number of Participants | Average Age | Sex |
|---|---|---|---|---|
| NC | 1344 | 123 | 69 ± 7.96 | F |
| MCI | 1008 | 92 | 75 ± 7.37 | F |
| AD | 1131 | 105 | 72 ± 6.22 | F |

NC: Normal cognition, MCI: Mild cognitive impairment, AD: Alzheimer's disease.

Lastly, we used mel-frequency cepstral coefficients (MFCCs) [29], features that take into account characteristics of the human auditory system. Unlike mel spectrograms, MFCCs involve further processing of the mel-scaled spectrum through cepstral analysis, providing a more compact and decorrelated representation of the speech signal. This property renders MFCCs effective features in traditional voice recognition systems, resulting in enhanced efficiency in subsequent machine learning tasks by reducing the correlation between the features.

## Models

Various AI models were used for cognitive disorder classification using audio data: 1D CNN, AST, Wav2Vec 2.0, AlexNet, DenseNet, Inception, ResNet, SqueezeNet, and VGG-19. The 1D CNN model (Fig 1), designed for audio processing in this study, comprised an initial Conv1D layer with a kernel size of 1 to process the input spectrogram (linear), mel-spectrogram, or MFCC.

This approach was employed to evaluate the model's performance across different input features and to establish a baseline for comparison. The 1D CNN was selected as a simple yet flexible model to investigate the impact of input feature choice. This layer was followed by six blocks of Conv1D layers, each with a kernel size of 3 and 64 channels. These

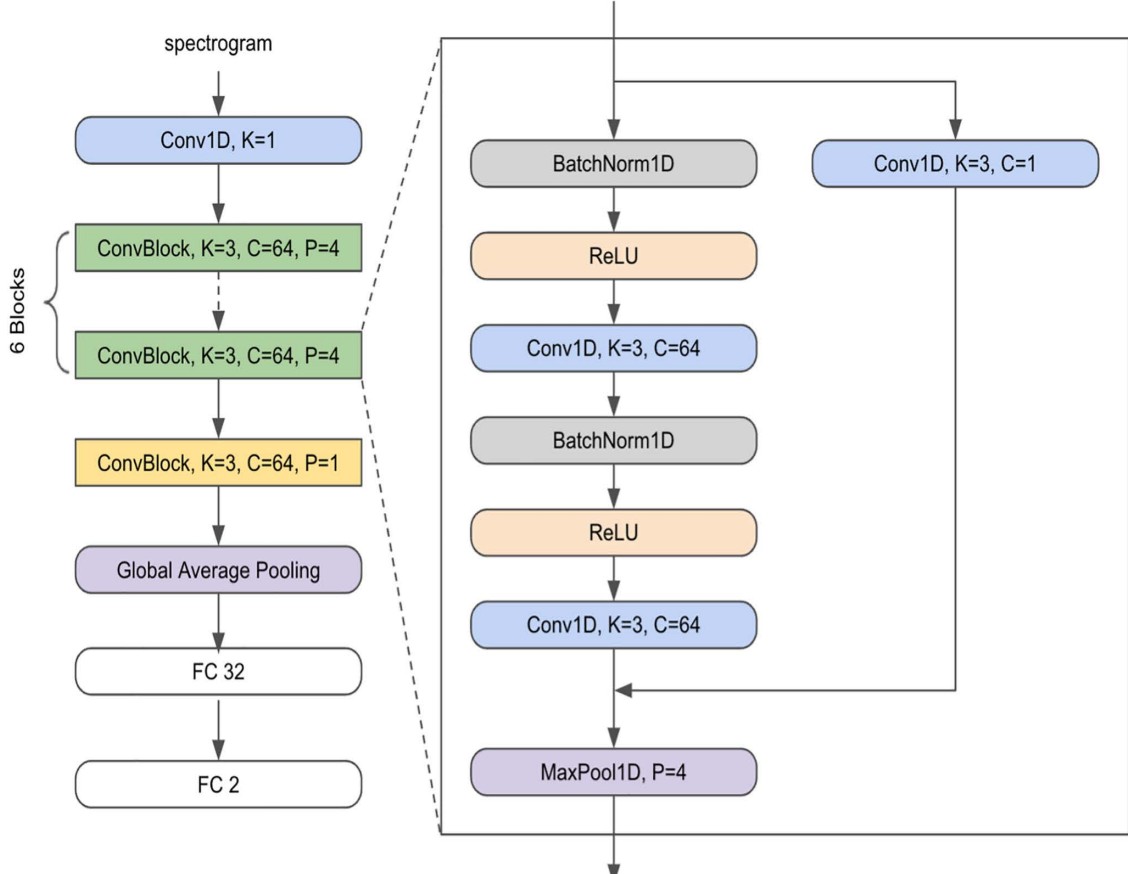

**Fig 1. Architecture of the 1D CNN (1-Dimensional Convolutional Neural Network) model.** The diagram illustrates the sequential transformation of raw audio features through feature extraction layers down to the final classification logic, highlighting its capability to process sequential time-series acoustic data.

blocks also included padding and stride parameters, with several blocks employing a stride of 4 for down-sampling. The sequence passed through a global average pooling layer that was further connected to two fully connected layers (FC), the first having 32 nodes and the second having two nodes, for binary classification. Batch normalization and rectified linear unit (ReLU) activation functions were applied throughout the network for stable training and non-linearity. The model also incorporated a skip connection that bypassed one of the Conv1D layers, followed by max pooling, which likely helped capture both the detailed and abstract features within the audio data.

Other architectures, including pre-trained CNNs and Transformer-based models, were exclusively trained on mel-spectrograms, as this representation aligns well with their design and optimization for 2D image-like inputs.

The AST model, representing a variant of the ViT model for audio data, accepts spectrogram images as inputs and applies them to audio classification tasks. The pretrained model used for learning in this study was initially trained on AudioSet data (https://paperswithcode.com/dataset/audioset), where it learned to classify sounds into 527 classes. Fig 2 illustrates that the input to the AST model was a mel-spectrogram, which underwent an overlapped patch split operation, effectively dividing the spectrogram into numerous segments.

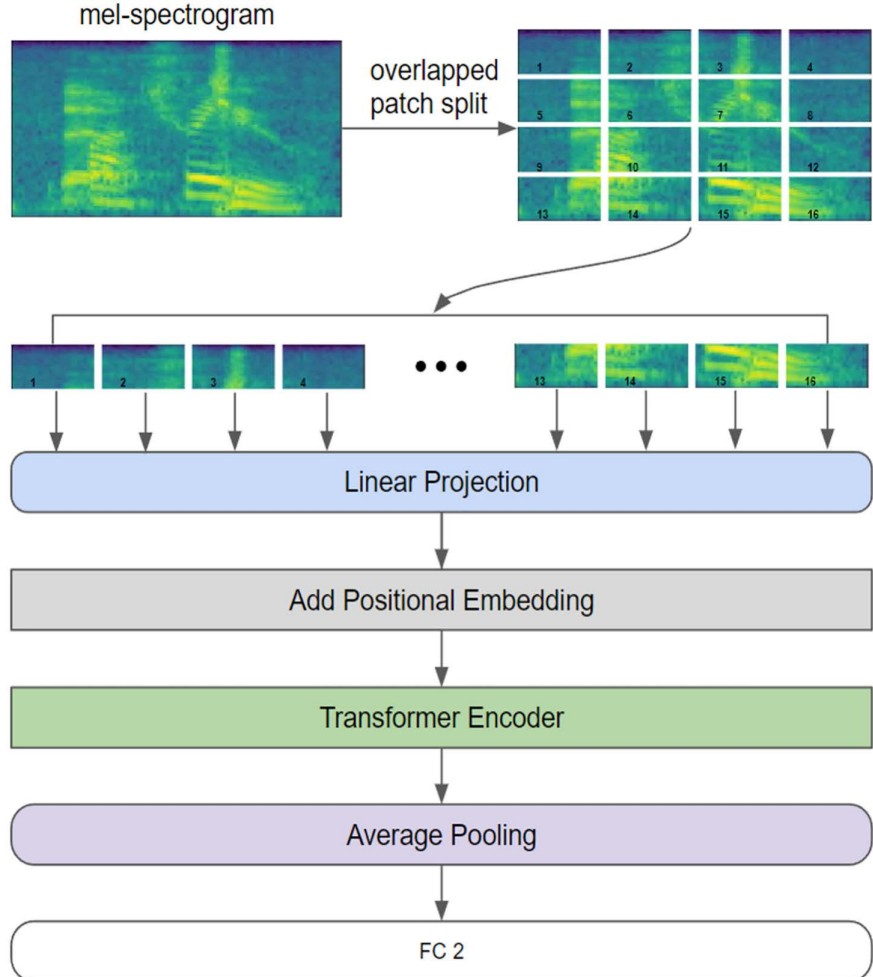

**Fig 2. Architecture of the AST (Audio Spectrogram Transformer) model.** This schematic details how 2D audio spectrograms are divided into localized patches and processed through transformer encoder blocks utilizing self-attention mechanisms to learn global acoustic context.

These segments were then individually subjected to a linear projection to transform them into a sequence of vectors suitable for processing by the transformer architecture. Subsequently, positional embeddings were added to these vectors to retain the sequential information crucial for the transformer encoder. This encoder processed the sequence by embedding contextual information into each vector. Next, an average pooling layer consolidated the output into a single vector, capturing the essence of the input features. Finally, this pooled vector was passed through two FCs, typically employed in binary classification tasks. This architecture allowed the model to learn from complex input sequences by leveraging the self-attention mechanism inherent in the transformer encoder.

The Wav2Vec 2.0 model used in this study featured a multi-layered architecture beginning with a waveform input (Fig 3). This input was first processed by a CNN layer, which encoded the raw audio data into a feature-rich, compressed form. Subsequently, a transformer encoder was used to process these features, capturing the temporal dynamics and relationships within the audio sequence.

The output from the encoder was then aggregated through an average pooling layer, which condensed the information into a single representative vector. Finally, this pooled vector was forwarded to two FCs, typically configured for binary output, which could be used for tasks such as speech recognition or audio classification. This architecture combined the localized feature extraction capabilities of CNNs with the global contextual understanding of transformer encoders to effectively process audio data; we used the XLSR-53, a large-scale model pretrained on 53

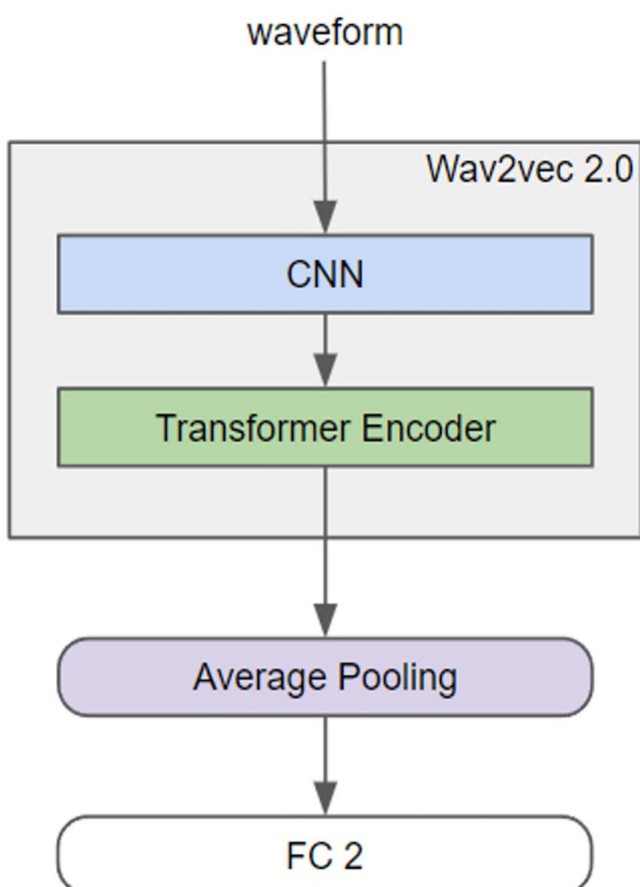

**Fig 3. Architecture of the Wav2Vec 2.0 model.** The figure demonstrates the self-supervised learning pipeline, showcasing the initial encoding of raw speech waveforms via a CNN block followed by deep contextualized representation learning within a robust transformer network.

languages by Facebook [30]. To enhance the ability of this model to extract features from older voices, we further trained it using publicly available elderly voice data. Subsequently, we used the pretrained Wav2Vec 2.0 model to extract voice embeddings and added a classifier to determine the presence of cognitive disorders, thereby constructing our classification model.

The following models were pretrained using the ImageNet dataset (https://www.image-net.org/). We fine-tuned all pretrained CNN models, including VGG-19, on mel-spectrogram data. The fine-tuning process involved replacing the final fully connected layers with task-specific layers, followed by retraining the models on the dataset used in this study.

**AlexNet** [31] was composed of five convolutional layers and three FC layers. Non-linearity was introduced using the ReLU activation function. Dropout was applied to prevent overfitting during training. The five convolutional layers extracted features from images, whereas the three FC layers were used for classification. Additionally, the model incorporated the ReLU activation function, overlapping max pooling, and dropout to mitigate overfitting. AlexNet accepted images sized 227 × 227 pixels as input. For audio analysis, mel-spectrogram images were used as training data. As a model pretrained on diverse images, AlexNet could effectively extract features useful for image processing tasks.

**DenseNet** [32] featured dense connections, and the features generated in each layer were forwarded to all subsequent layers. This configuration facilitated effective learning and feature reuse with a limited number of parameters. This model took 224 × 224 pixel images as input. For audio analysis, DenseNet used mel-spectrogram images as training data.

The **Inception model** incorporated inception modules, using kernels of various sizes in parallel for the simultaneous extraction of multi-scale features. It accepted mel-spectrogram images as the input for audio analysis and effectively captured audio patterns of various sizes through the inception module.

**ResNet** [33] passed the input directly to deeper layers through residual connections. This structure facilitated learning even as the network became deeper. ResNet-50 accepted images sized 224 × 224 pixels as the input. This model used mel-spectrogram images as training data.

**SqueezeNet** aimed at maintaining high performance while reducing the number of parameters. To this end, it used a special structure named the "Fire" module for effectively extracting features with a small number of parameters. The model was fine-tuned using mel-spectrogram images.

**VGG-19** [34] was composed of 16 convolutional layers and three FC layers. All convolutional layers used small 3 × 3 filters, along with max pooling to reduce image size. VGG-19 learned complex features through its deep layers. It took images sized 224 × 224 pixels as the input.

### Training configuration

All deep learning models were trained with a unified set of core hyperparameters to ensure fair comparison. The AdamW optimizer (Adam with decoupled weight decay regularization) was employed with a base learning rate of $5 \times 10^{-5}$. A cosine annealing warm restarts learning rate scheduler was applied ($T_0 = 20{,}000$ steps, T_mult = 2, η_min = $1 \times 10^{-5}$) to facilitate stable convergence. The batch size was set to 1 per GPU step with gradient accumulation over 16 steps, yielding an effective batch size of 16. Mixed-precision training was utilized via PyTorch's automatic mixed precision (autocast and GradScaler) to reduce memory consumption and accelerate computation. Gradient clipping with an L2 norm constraint of 2.0 was applied to prevent gradient explosion. The loss function used was binary cross-entropy (BCE) loss. An early stopping strategy based on validation loss was implemented with a patience of 10 epochs without improvement. All experiments were conducted on a system equipped with an NVIDIA Geforce RTX 3090 GPU (24 GB memory) and a CPU with 32 GB of RAM. The GPU memory requirements varied by model architecture: Wav2Vec 2.0 required approximately 23 GB, the AST (Transformer) required approximately 20 GB, and the 1D CNN and image-based pre-trained models (AlexNet, VGG-19, ResNet, DenseNet, Inception, SqueezeNet) each required approximately 10 GB of GPU memory.

## Model evaluation

The classifier performances were assessed using accuracy, recall, precision, and F1-score.

$$Accuracy = (TP + TN) / (TP + TN + FP + FN)$$

$$Precision = TP / (TP + FP)$$

$$Recall\ (Sensitivity) = TP / (TP + FN)$$

$$F1 - Score = 2 \times (Precision \times Recall) / (Precision + Recall)$$

Where TP, TN, FP, and FN represent True Positives, True Negatives, False Positives, and False Negatives, respectively.

To ensure unbiased evaluations and improve model generalization, a five-fold cross-validation procedure was implemented, wherein each fold served once as validation data, while the remaining folds served as training data in each learning process. The outcome was calculated as the average of the results from the five-fold dataset. To mitigate potential bias and improve the model's ability to classify cognitive disorders, the data of each individual was confined to a single fold in the cross-validation procedure, ensuring the model learned from disorder-related features rather than individual variations.

Specifically, the fold assignment was stratified by cognitive status label to ensure that each fold maintained a similar proportion of participants from each diagnostic group. An under-sampling strategy was employed to balance class representation: the number of participants per class within each fold was adjusted to match the minority class, with gradual per-participant data point adjustment to achieve equitable class distributions. This participant-level separation guaranteed that all recordings from a given individual appeared exclusively in either the training or validation set within any given fold, thereby eliminating data leakage and ensuring that the model generalized based on disorder-related acoustic features rather than memorizing individual speaker characteristics. Each binary classification task (NC vs. MCI and NC vs. AD) was evaluated independently through this cross-validation framework. To assess the statistical significance of performance differences between models, pairwise proportion z-tests were conducted comparing each model's classification accuracy against that of Wav2Vec 2.0 (the best-performing model). Since each participant was assigned to exactly one test fold in the five-fold cross-validation, the aggregated accuracy across all folds represents the overall proportion of correctly classified participants. Wilson score intervals were computed to obtain 95% confidence intervals (CIs) for each model's accuracy. To control the family-wise error rate across multiple pairwise comparisons, Holm-Bonferroni correction was applied to the resulting p-values. Cohen's h effect size was computed for each comparison to quantify the magnitude of performance differences. A significance threshold of $\alpha = 0.05$ was adopted.

## Results

Performance evaluations were conducted on 11 models. The results of these evaluations on unseen test data are summarized in Table 2. The models were divided into two categories: those classifying MCI and NC and those classifying AD and NC.

In the MCI vs. NC classification (Table 2, left column), Wav2Vec 2.0 demonstrated the highest accuracy of 0.74 (95% CI [0.678, 0.794]), with a precision of 0.83, a recall of 0.74, and an F1-score of 0.72. The other models showed varying degrees of performance, with SqueezeNet displaying the lowest accuracy of 0.48 (95% CI [0.414, 0.547]).

**Table 2. Performance metrics and statistical comparisons for MCI and AD classification models.**

| Model | MCI vs NC (n=215) | | | | | AD vs NC (n=228) | | | | | |
|---|---|---|---|---|---|---|---|---|---|---|---|
| | Accuracy | 95% CI (MCI) | Precision | Recall | F1-score | Accuracy | 95% CI (AD) | Precision | Recall | F1-score | p† (AD vs NC) |
| Wav2Vec 2.0 | 0.74 | [0.678 0.794] | 0.83 | 0.74 | 0.72 | 0.83 | [0.776 0.873] | 0.83 | 0.83 | 0.83 | ref. |
| 1D CNN | | | | | | | | | | | |
| Spectrogram | 0.65 | [0.584 0.711] | 0.68 | 0.65 | 0.63 | 0.71 | [0.648 0.765] | 0.72 | 0.71 | 0.71 | 0.014* |
| Mel-Spectrogram | 0.61 | [0.543 0.673] | 0.63 | 0.61 | 0.58 | 0.63 | [0.566 0.690] | 0.63 | 0.63 | 0.63 | <0.001*** |
| MFCC | 0.66 | [0.594 0.720] | 0.66 | 0.66 | 0.66 | 0.66 | [0.596 0.718] | 0.66 | 0.66 | 0.66 | <0.001*** |
| AST | 0.56 | [0.493 0.625] | 0.67 | 0.56 | 0.49 | 0.52 | [0.455 0.584] | 0.51 | 0.52 | 0.49 | <0.001*** |
| Alexnet | 0.61 | [0.543 0.673] | 0.61 | 0.61 | 0.60 | 0.72 | [0.658 0.774] | 0.75 | 0.72 | 0.72 | 0.025* |
| Inception | 0.52 | [0.453 0.586] | 0.52 | 0.52 | 0.52 | 0.67 | [0.607 0.728] | 0.67 | 0.66 | 0.66 | <0.001*** |
| VGG-19 | 0.56 | [0.493 0.625] | 0.58 | 0.57 | 0.55 | 0.74 | [0.679 0.793] | 0.75 | 0.74 | 0.73 | 0.058 |
| ResNet | 0.55 | [0.483 0.615] | 0.55 | 0.55 | 0.54 | 0.73 | [0.669 0.783] | 0.74 | 0.73 | 0.73 | 0.040* |
| DenseNet | 0.65 | [0.584 0.711] | 0.65 | 0.65 | 0.65 | 0.78 | [0.722 0.829] | 0.79 | 0.78 | 0.78 | 0.356 |
| SqueezeNet | 0.48 | [0.414 0.547] | 0.49 | 0.48 | 0.45 | 0.80 | [0.743 0.847] | 0.8 | 0.79 | 0.79 | 0.409 |

NC, Normal Cognition, MCI: Mild Cognitive Impairment, AD: Alzheimer's disease; CNN, convolutional neural network; AST, audio spectrogram transformer; VGG-19, Visual Geometry Group-19; CI, confidence interval. † Holm-Bonferroni corrected p-values from pairwise proportion z-tests comparing each model's accuracy against Wav2Vec 2.0 in the NC vs. AD task. Wilson score 95% confidence intervals are shown in brackets.

* p<0.05, ** p<0.01, *** p<0.001.

In the AD vs. NC classification (Table 2, right column), the Wav2Vec 2.0 model again demonstrated the best performance with an accuracy of 0.83 (95% CI [0.776, 0.873]), along with precision, recall, and F1-score values of 0.83. The performances of the other models varied in this task as well, with the AST model presenting the lowest accuracy of 0.52 (95% CI [0.455, 0.584]).

Pairwise proportion z-tests with Holm-Bonferroni correction confirmed that Wav2Vec 2.0 significantly outperformed the majority of competing models. In the NC vs. AD task, 7 of 10 pairwise comparisons reached statistical significance (corrected $p < 0.05$): 1D CNN Spectrogram ($p = 0.014$), Mel-Spectrogram ($p < 0.001$), MFCC ($p < 0.001$), AST ($p < 0.001$), AlexNet ($p = 0.025$), Inception ($p < 0.001$), and ResNet ($p = 0.040$). Three models did not differ significantly from Wav2Vec 2.0: VGG-19 ($p = 0.058$), DenseNet ($p = 0.356$), and SqueezeNet ($p = 0.409$). Cohen's h effect sizes ranged from 0.08 (SqueezeNet) to 0.69 (AST), indicating small to medium practical differences. Wilson score 95% confidence intervals for all model accuracies are reported in Table 2.

Fig 4 presents the confusion matrices for the Wav2Vec 2.0 and 1D CNN (Spectrogram) models based on aggregated predictions across all five cross-validation folds. For the Wav2Vec 2.0 model, the NC vs. MCI classification (Fig 4A) correctly identified 69 out of 92 MCI cases and 90 out of 123 NC cases, yielding a sensitivity of 0.75 and a specificity of 0.73. In the NC vs. AD task (Fig 4B), the model achieved higher performance with a sensitivity of 0.83 (87/105 AD) and a specificity of 0.83 (102/123 NC).

In contrast, the 1D CNN (Spectrogram) model showed lower classification accuracy. In the NC vs. MCI task (Fig 4C), it correctly identified 60 MCI and 80 NC cases (sensitivity 0.65, specificity 0.65). Similarly, for the NC vs. AD classification (Fig 4D), the sensitivity and specificity were both 0.71, with 75 of 105 AD and 87 of 123 NC cases correctly predicted.

## Discussion

The use of three input types (spectrogram, mel-spectrogram, and MFCC) for the 1D CNN model was specifically intended to benchmark its performance under varying input conditions. This was done to provide insights into the

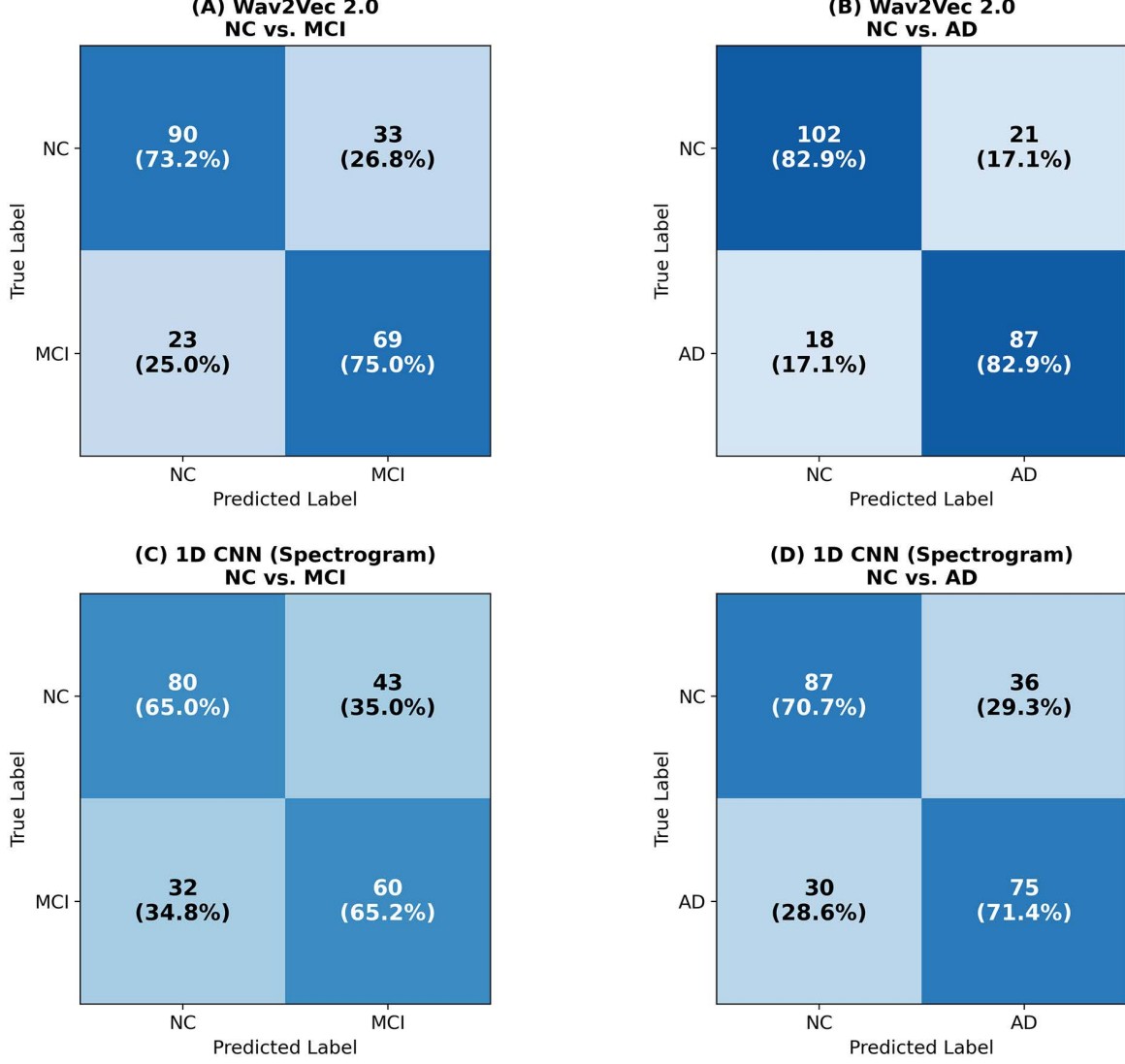

**Fig 4. Confusion matrices for the Wav2Vec 2.0 and 1D CNN (Spectrogram) models across binary classification tasks. (A)** Wav2Vec 2.0 for NC vs. MCI. **(B)** Wav2Vec 2.0 for NC vs. AD. **(C)** 1D CNN (Spectrogram) for NC vs. MCI. **(D)** 1D CNN (Spectrogram) for NC vs. AD. Values represent aggregated predictions across all five cross-validation folds. NC, Normal Cognition; MCI, Mild Cognitive Impairment; AD, Alzheimer's Disease.

model's ability to adapt to different feature representations. For pre-trained CNN and Transformer-based models, only mel-spectrograms were used to maximize compatibility with their 2D input structure and computational efficiency. This design decision reflects the tailored strengths of each model type and ensures fair comparisons within their respective capabilities.

The inclusion of six CNN architectures allowed for a thorough evaluation of commonly used models in the field of deep learning. Each architecture offers unique advantages: AlexNet for its simplicity and efficiency, ResNet for its use of residual connections to combat vanishing gradients, DenseNet for its dense connectivity enabling feature reuse, and so on. This comparative approach highlights the strengths and limitations of each architecture when applied to the task of cognitive impairment detection using speech data.

Among the considered models, the Wav2Vec 2.0 model emerged as superior in classifying MCI and NC as well as AD and NC. This observation highlighted the potential strengths of this model in identifying the presence of cognitive disorders through detailed speech analysis.

The exceptional performance of Wav2Vec 2.0 was attributed to its unique learning approach [15,23]. Unlike other models, it was pretrained using large-scale voice data, exposing it to a diverse range of voice patterns and speech nuances. Consequently, this model could more effectively handle complex features and subtle patterns of speech than other models. This ability is particularly beneficial when dealing with cognitive disorders, where slight deviations in speech can provide crucial insights into the patient's condition.

Models based on 1D CNN, DenseNet, or SqueezeNet did not achieve the same level of performance as Wav2Vec 2.0. This discrepancy in performance was likely attributable to the limited ability of these models to capture complex features from audio signals. Wav2Vec 2.0, with a transformer-based architecture, could effectively handle more intricate patterns and extract detailed features from speech, essential for tasks such as speech recognition and sentiment analysis from audio data [15]. These findings align with existing research demonstrating the superior capabilities of Wav2Vec 2.0 in capturing nuances from audio inputs, which may be challenging to achieve for models that simplify the audio signal, such as those using MFCC or mel-spectrogram features alone. Furthermore, integrating features such as speaker identity and phone sequence embeddings can further enhance the performance of models such as Wav2Vec 2.0, which are designed to handle complex speech representations [15,23,35].

The notably lower accuracy of models relying on simplified feature representations (e.g., 1D CNN with MFCC achieving 0.66 in both tasks) compared to Wav2Vec 2.0 underscores that manual feature engineering alone is insufficient for capturing the complex acoustic patterns associated with cognitive impairment. This observation supports the intrinsic diagnostic value of automated representation learning, particularly the self-supervised approach employed by Wav2Vec 2.0, which can discover latent speech features without explicit feature design.

Some models used mel-spectrograms as input, as they were originally designed for processing such data. The preference for mel-spectrograms over MFCCs in most models was based on dimensional compatibility with 2D CNN architectures typically found in image models. These image models typically expect input shapes with dimensions of [244, 244] or [299, 299]. Mel-spectrograms fit more closely with this dimension requirement, whereas MFCCs, with a considerably smaller dimensionality of 13, would necessitate a nearly 20-fold increase to match the expected input size, potentially leading to inefficient computational scaling. This computational inefficiency, alongside considerations of training time, led to the prioritization of mel-spectrograms over MFCCs for models other than 1D CNN.

Another notable aspect is the difference in the performance of models classifying AD and NC versus those classifying MCI and NC. Specifically, the models classifying AD and NC generally outperformed those classifying MCI and NC. This observation may be attributable to the cognitive impairment symptoms associated with MCI and AD. MCI generally represents an intermediate stage between the cognitive decline expected with aging and the more serious decline associated with dementia, particularly in AD [3]. Patients with MCI have a significantly increased risk of developing AD, with an annual rate of 15.7% [4], although MCI does not always lead to AD. The challenge in predicting MCI compared with AD is that MCI can remain stable or even revert to NC [5,6], whereas AD has a progressive nature. Therefore, models generally achieved higher accuracy in AD classification [36], as AD produces more distinct speech alterations, whereas the subtle and heterogeneous nature of MCI made accurate classification inherently more challenging. The complexity of features captured by a model like Wav2Vec 2.0 could lead to better performance in identifying AD [36,37]. These aspects could potentially explain why models distinguishing MCI and NC generally underperformed compared with those distinguishing AD and NC.

The objective of this research was to establish a reliable, non-invasive diagnostic tool using vocal analysis, thereby promoting the early detection and management of cognitive disorders. Our evaluation of various model training methods provided valuable insights into the strengths and limitations of each approach, facilitating informed decision-making for researchers and practitioners in the field of cognitive impairment prediction.

While the focus of this study was on binary classifications (NC vs. MCI and NC vs. AD), we recognize the importance of multi-class classification (NC vs. MCI vs. AD) for capturing the nuanced progression of cognitive impairment. Preliminary experiments with the three-class task using Wav2Vec 2.0 did not yield meaningful results, likely due to the overlap in speech features between MCI and the other two classes. Future research will explore strategies to enhance multi-class classification performance, including advanced pretraining, fine-tuning on task-specific datasets, and incorporating additional speech features to better distinguish between cognitive states.

Several methodological limitations of this study must be addressed to appropriately contextualize our findings. First, regarding dataset size, while our cross-validation strategy rigorously ensured internal robustness, the overall participant count (N = 320) remains relatively limited for training complex transformer architectures like Wav2Vec 2.0. A larger cohort is necessary to fully exploit model capacity and prevent subtle overfitting. Second, we acknowledge a critical gender bias within our sample. We initially prioritized the female cohort solely to prevent extreme class imbalance, given the limited availability of male recordings across target cognitive stages in the AI Hub repository. We acknowledge that relying on a female-only dataset limits the generalizability of our findings to male populations, because gender-specific vocal characteristics such as fundamental frequency and formants significantly affect acoustic feature representations. Consequently, future studies should incorporate gender-balanced cohorts to ensure comprehensive and equitable diagnostic applicability. Finally, the models were trained exclusively on native Korean speakers. Because phonetics, articulation rates, and pausing intervals intrinsically vary across languages, the current findings may not generalize directly to other linguistic populations. Future multinational-scale validations are required.

A further direction for future research involves the analysis of the 11 individual speech tasks. Our current approach combined these tasks to aggregate continuous acoustic features. However, different verbal prompts (e.g., complex storytelling versus simple repetition) engage distinct cognitive faculties, potentially yielding different biomarkers associated with neurodegeneration. In a future study, we will test this hypothesis, emphasizing that isolating individual cognitive tasks to identify the most informative speech biomarkers represents an important direction for streamlining diagnostic data collection.

## Conclusions

In conclusion, our findings underscored the importance of choosing the appropriate model and features for cognitive disorder classification. We highlight the superior performance of the Wav2Vec 2.0 model, limitations of models using simplified features, and influence of the severity of cognitive symptoms on model performance. Future studies could further explore these aspects to develop more accurate and effective models for diagnosing and monitoring cognitive disorders.

## Acknowledgments

The authors thank all participants who contributed to this study and the research staff who assisted with data collection and analysis.

## Author contributions

**Conceptualization:** Gihyun Yun, Young Chul Youn.

**Data curation:** Hunboc Lee.

**Funding acquisition:** Gihyun Yun, Young Chul Youn.

**Methodology:** SangYun Kim.

**Supervision:** SangYun Kim, Young Chul Youn.

**Writing – original draft:** Minsoo Kim, Young Chul Youn.

**Writing – review & editing:** Hyunjoo Choi, YongSoo Shim, Nayoung Ryoo, Ho Tae Jeong, Gihyun Yun, Hunboc Lee, SangYun Kim.

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
