## [Decision Letter · Decision Letter 0]

13 Feb 2026

PONE-D-25-44894Prediction of Cognitive Impairment through Speech Data Analysis: A Comparative Evaluation of Deep Learning ModelsPLOS One

Dear Dr. Youn,

Thank you for submitting your manuscript to PLOS ONE. After careful consideration, we feel that it has merit but does not fully meet PLOS ONE’s publication criteria as it currently stands. Therefore, we invite you to submit a revised version of the manuscript that addresses the points raised during the review process.

Dear authors,

Thank you for your submission. At this moment, we couldn't accept your article. Would you please consider below comments, revise it and submit for further review?

We look forward to receiving your revised manuscript.

Kind regards,

Farman Ullah

Academic Editor

PLOS One

**Journal Requirements:**

1. When submitting your revision, we need you to address these additional requirements. Please ensure that your manuscript meets PLOS ONE's style requirements, including those for file naming. The PLOS ONE style templates can be found at https://journals.plos.org/plosone/s/file?id=wjVg/PLOSOne_formatting_sample_main_body.pdf and https://journals.plos.org/plosone/s/file?id=ba62/PLOSOne_formatting_sample_title_authors_affiliations.pdf 2. Please note that PLOS One has specific guidelines on code sharing for submissions in which author-generated code underpins the findings in the manuscript. In these cases, we expect all author-generated code to be made available without restrictions upon publication of the work. Please review our guidelines at https://journals.plos.org/plosone/s/materials-and-software-sharing#loc-sharing-code and ensure that your code is shared in a way that follows best practice and facilitates reproducibility and reuse. 3. Please include your tables as part of your main manuscript and remove the individual files. Please note that supplementary tables (should remain/ be uploaded) as separate "supporting information" files. 4. We note that the grant information you provided in the ‘Funding Information’ and ‘Financial Disclosure’ sections do not match.  When you resubmit, please ensure that you provide the correct grant numbers for the awards you received for your study in the ‘Funding Information’ section. 5. Thank you for stating in your Funding Statement: G.Y. was supported by grants from the Ministry of SMEs and Startups (Project Number: S3079103). Y.C.Y acknowledges funding from the Cooperative Research Program for Agriculture Science and Technology Development Rural Development Administration (Project No. PJ01712403). No, the sponsors/funders had no role in the study design, data collection and analysis, decision to publish, or preparation of the manuscript. The research was conducted independently by the authors. Please provide an amended statement that declares *all* the funding or sources of support (whether external or internal to your organization) received during this study, as detailed online in our guide for authors at http://journals.plos.org/plosone/s/submit-now. Please also include the statement “There was no additional external funding received for this study.” in your updated Funding Statement. Please include your amended Funding Statement within your cover letter. We will change the online submission form on your behalf. 6. Thank you for uploading your study's underlying data set. Unfortunately, the repository you have noted in your Data Availability statement does not qualify as an acceptable data repository according to PLOS's standards. At this time, please upload the minimal data set necessary to replicate your study's findings to a stable, public repository (such as figshare or Dryad) and provide us with the relevant URLs, DOIs, or accession numbers that may be used to access these data. For a list of recommended repositories and additional information on PLOS standards for data deposition, please see https://journals.plos.org/plosone/s/recommended-repositories. 7. PLOS requires an ORCID iD for the corresponding author in Editorial Manager on papers submitted after December 6th, 2016. Please ensure that you have an ORCID iD and that it is validated in Editorial Manager. To do this, go to ‘Update my Information’ (in the upper left-hand corner of the main menu), and click on the Fetch/Validate link next to the ORCID field. This will take you to the ORCID site and allow you to create a new iD or authenticate a pre-existing iD in Editorial Manager. 8. Your ethics statement should only appear in the Methods section of your manuscript. If your ethics statement is written in any section besides the Methods, please move it to the Methods section and delete it from any other section. Please ensure that your ethics statement is included in your manuscript, as the ethics statement entered into the online submission form will not be published alongside your manuscript. 9. If the reviewer comments include a recommendation to cite specific previously published works, please review and evaluate these publications to determine whether they are relevant and should be cited. There is no requirement to cite these works unless the editor has indicated otherwise.

**Additional Editor Comments:**

Dear authors,

Thank you for your submission. At this moment, we couldn't accept your article. Would you please consider below comments, revise it and submit for further review?

Reviewers' comments:

Reviewer's Responses to Questions

**Comments to the Author**

1. Is the manuscript technically sound, and do the data support the conclusions?

Reviewer #1: Yes

Reviewer #2: Yes

2. Has the statistical analysis been performed appropriately and rigorously? 

Reviewer #1: No

Reviewer #2: Yes

3. Have the authors made all data underlying the findings in their manuscript fully available?

Reviewer #1: Yes

Reviewer #2: Yes

4. Is the manuscript presented in an intelligible fashion and written in standard English?

Reviewer #1: No

Reviewer #2: Yes

5. Review Comments to the Author

**Reviewer #1:** This manuscript explores an important and emerging field of speech-based detection of cognitive impairment using deep learning models. The comparative approach across multiple architectures (1D CNN, AST, Wav2Vec 2.0, etc.) is valuable and well-aligned with the journal’s scope. However, the study requires major revisions to improve methodological transparency, statistical rigor, interpretability, and presentation clarity before it can be considered for publication.

1. Clarify dataset composition, total participants, and inclusion of all 11 speech tasks.

2. Address data imbalance (e.g., female-majority dataset) and its effect on generalization.

3. Describe preprocessing steps such as noise reduction, silence trimming, and normalization.

4. Specify details of data splits and confirm participant-level separation in cross-validation.

5. Provide hyperparameter tuning details (learning rate, epochs, optimizer, batch size).

6. Add statistical tests to verify significance of model performance differences.

7. Present confusion matrices or performance plots for key models.

8. Compare with simpler baseline models (e.g., SVM or logistic regression with MFCC).

9. Expand limitations regarding dataset size, gender bias, and language specificity.

10. Add DOI: 10.1109/OJCS.2025.3580570, this study’s use of 1D-to-2D transformations (CWT, STFT) and CNN-based modelling parallels the present work’s spectrogram approach, reinforcing its methodological validity and connecting speech-based cognitive analysis to broader applications in noninvasively biomedical signal processing.

12. Add model complexity metrics (parameters, training time, memory).

13. Strengthen abstract by summarizing dataset size and highlighting key contributions.

14. Improve figure captions and ensure consistent terminology across sections.

15. Edit text for conciseness, tense consistency, and readability for non-technical readers.

**Reviewer #2:** This study addresses the critical need for no-invasive diagnostic tools for cognitive decline, by evaluating multiple deep learning architectures applied to dataset from “Cognitive Impairment Diagnosis Voice/Conversation”. Using this dataset the study compares the efficacy of 1D CNN, AST and Wav2Vec2.0 in classifying binary tasks: normal cognition (NC) vs. mild cognitive impairment (MCI) and NC vs. Alzheimer’s disease (AD). The paper concludes that Wav2Vec2.0 outperforms other architectures, particularly in the NC vs. AD classification task.

The primary strength of this work lies in systematic, side by side evaluation of disparate modeling approaches (eg. VGG-19, ResNet, transformer-based models) to audio models providing a clear hierarchy of effectiveness. In particular, the results reveal that self-supervised speech representation model such as Wav2Vec2.0 is better suited for cognitive changes in speech.

I had a few comments that I believe would improve the paper and its applicability.

1.This study prioritizes female voices due to lower volume of male recordings. Although this choice is understandable, it raises concern about generalizability of the results. It would have been interesting to see how sex-specific vocal characteristics might influence model performance and whether conclusions are expected to hold for equally mixed/male-dominated cohorts.

2.If I could make out correctly, the results aggregate the performance across the 11 different tasks (eg. Sentence repetition, language fluency). In the limiting case what would be the effect of analyzing these individually? Some tasks (eg. storytelling) are biologically more relevant for cognitive decline than other (Image description). I was wondering which specific tasks account for higher Wav2Vec2.0. This in turn could streamline future data collections.

3.Finally, the model performance is reported using across five-fold cross-validation. Given relatively modest performance differences between some models, reporting statistical significance testing/confidence interval would help clarify the robustness.

Overall, this manuscript presents a well-executed study using speech dataset and deep learning techniques. The comparative analysis is valuable contribution to the field of digital biomarker. I believe addressing above points would strengthen the paper.

6. PLOS authors have the option to publish the peer review history of their article (what does this mean?). If published, this will include your full peer review and any attached files.

Reviewer #1: No

Reviewer #2: No

---

## [Author Response · Author response to Decision Letter 1]

19 Mar 2026

Journal Requirements:

Response: Thank you for your valuable feedback on our manuscript. We have carefully revised the manuscript according to the reviewers' comments. The specific changes are as follows:

Manuscript Title: The title has been updated to Sentence case as requested.

Author Information: Academic degrees (MD, PhD) have been removed, and the corresponding author's information has been simplified to include only the email address and initials.

Section Headings: We have adjusted the font sizes (18pt and 16pt) and applied Sentence case formatting to all section titles.

Figure Captions: All figure captions have been repositioned to appear immediately following their first citation within the text.

We hope these revisions meet the journal's requirements.

Response: While we fully support PLOS ONE's software sharing guidelines, our author-generated code (utilized for audio preprocessing and deep learning model training) was developed in collaboration with a commercial partner. Due to the inclusion of proprietary company know-how and intellectual property restrictions, we are unable to make the source code publicly available in an unrestricted repository. However, to facilitate the rigorous review process and ensure methodological transparency, we are fully prepared to provide the code confidentially to the reviewers.

Private link: https://figshare.com/s/3d395292a035369d4bac

3. Please include your tables as part of your main manuscript and remove the individual files. Please note that supplementary tables (should remain/ be uploaded) as separate "supporting information" files.

Response: All tables have been directly inserted into the main manuscript text rather than uploaded as separate individual files.

Response: We sincerely apologize for the initial discrepancy. As requested, we have carefully amended the Funding Statement within our Cover Letter to explicitly list all precise sources of support. The updated statement now explicitly includes the Tech Incubator Program for Startup Korea (TIPS), the Korea Technology and Information Promotion Agency for SMEs (TIPA), and the Ministry of SMEs and Startups (MSS).

“This work was supported by the Tech Incubator Program for Startup Korea (TIPS) through the Korea Technology and Information Promotion Agency for SMEs (TIPA) funded by the Ministry of SMEs and Startups (MSS, Republic of Korea) (Project No. S3079103) and the Cooperative Research Program for Agriculture Science and Technology Development Rural Development Administration (Project No. PJ01712403). There was no additional external funding received for this study.”

5. Thank you for stating in your Funding Statement:

G.Y. was supported by grants from the Ministry of SMEs and Startups (Project Number: S3079103). Y.C.Y acknowledges funding from the Cooperative Research Program for Agriculture Science and Technology Development Rural Development Administration (Project No. PJ01712403).

No, the sponsors/funders had no role in the study design, data collection and analysis, decision to publish, or preparation of the manuscript. The research was conducted independently by the authors.

Response: We have updated the funding statement in the manuscript and cover letter.

“Funding

This work was supported by the Tech Incubator Program for Startup Korea (TIPS) through the Korea Technology and Information Promotion Agency for SMEs (TIPA) funded by the Ministry of SMEs and Startups (MSS, Republic of Korea) (Project No. S3079103) and the Cooperative Research Program for Agriculture Science and Technology Development Rural Development Administration (Project No. PJ01712403). There was no additional external funding received for this study.”

6. Thank you for uploading your study's underlying data set. Unfortunately, the repository you have noted in your Data Availability statement does not qualify as an acceptable data repository according to PLOS's standards.

Response: We appreciate this important point. The original audio dataset used in this study is available to approved researchers through the Korean AI Hub Safe Zone (https://safezone.aihub.or.kr/). Due to participant privacy regulations, unrestricted public distribution of the original audio recordings is not possible. However, the data is openly accessible to any global researcher upon application for scholarly purposes. To reproduce our findings, interested scientists can formally access the precise identical dataset by registering at the AI Hub Safe Zone (https://safezone.aihub.or.kr/ ) and electronically agreeing to the ethical terms of use.

To comply with PLOS's data sharing standards, we have deposited the core source code necessary for reproducing our study's findings on figshare as a private item. A private sharing link has been provided to reviewers (https://figshare.com/s/3d395292a035369d4bac), The deposited code includes:

- Complete implementations of all 11 model architectures (1D CNN, AST, Wav2Vec 2.0, and 6 ImageNet-pretrained CNNs)

- The training pipeline (AdamW optimizer, cosine annealing warm restarts, mixed-precision training, early stopping)

- The audio preprocessing pipeline (Bessel bandpass filtering, spectrogram/mel-spectrogram/MFCC extraction)

- Evaluation utilities (classification reports, confusion matrices, k-fold cross-validation aggregation)

7. PLOS requires an ORCID iD for the corresponding author in Editorial Manager on papers submitted after December 6th, 2016. Please ensure that you have an ORCID iD and that it is validated in Editorial Manager. To do this, go to ‘Update my Information’ (in the upper left-hand corner of the main menu), and click on the Fetch/Validate link next to the ORCID field. This will take you to the ORCID site and allow you to create a new iD or authenticate a pre-existing iD in Editorial Manager.

Response: As your recommendation, My ORCID (0000-0002-2742-1759) have been authenticated and updated.

8. Your ethics statement should only appear in the Methods section of your manuscript. If your ethics statement is written in any section besides the Methods, please move it to the Methods section and delete it from any other section. Please ensure that your ethics statement is included in your manuscript, as the ethics statement entered into the online submission form will not be published alongside your manuscript.

Response: We have relocated the ethics statement straight to the 'Methods' section and deleted it from the 'Declarations' section at the end of the manuscript.

Additional Editor Comments:

Reviewers' comments:

Reviewer's Responses to Questions

Comments to the Author

1. Is the manuscript technically sound, and do the data support the conclusions?

Reviewer #1: Yes

Reviewer #2: Yes

2. Has the statistical analysis been performed appropriately and rigorously?

Reviewer #1: No

Reviewer #2: Yes

3. Have the authors made all data underlying the findings in their manuscript fully available?

Reviewer #1: Yes

Reviewer #2: Yes

4. Is the manuscript presented in an intelligible fashion and written in standard English?

Reviewer #1: No

Reviewer #2: Yes

5. Review Comments to the Author

Reviewer #1: This manuscript explores an important and emerging field of speech-based detection of cognitive impairment using deep learning models. The comparative approach across multiple architectures (1D CNN, AST, Wav2Vec 2.0, etc.) is valuable and well-aligned with the journal’s scope. However, the study requires major revisions to improve methodological transparency, statistical rigor, interpretability, and presentation clarity before it can be considered for publication.

1. Clarify dataset composition, total participants, and inclusion of all 11 speech tasks.

Response: We thank the reviewer for identifying this ambiguity. We have comprehensively revised the 'Methods - Dataset' section. We now explicitly state the absolute number of total participants analyzed, and we clarified that the audio recordings spanning all 11 heterogeneous speech tasks were aggregated and utilized collectively per participant to generate sufficient temporal data for deep learning algorithms.

“Dataset

The dataset used in this study was acquired from the "Cognitive Impairment Diagnosis Voice/Conversation" offline data on 07/09/2022, which can be accessed at the AI Hub Safe Zone (https://safezone.aihub.or.kr/). This dataset contains voice data from individuals with dementia and control groups, collected through the SpeechTaskSet application. This clinical app is structured to guide participants through 11 heterogeneous speech tasks, including sentence repetition (three tasks), image description (two tasks), language fluency (two tasks), calculation (one task), and storytelling (three tasks), producing an audio file of up to 1 min per task. Notably, the app captures only the voices of the participants, excluding any input from the instructors. To generate sufficient temporal data for deep learning algorithms, the audio recordings spanning all 11 speech tasks were aggregated and utilized collectively per participant.” … “The AI Hub dataset included 3,483 voice recordings from 320 female participants and 1,481 from 136 male participants. Owing to the lower volume of male recordings, this study prioritized the female voice dataset for model training to prevent extreme class imbalance. A total of 320 female participants were analyzed, including 105 with AD, 92 with MCI, and 123 cognitively normal controls (NC) (Table 1). The recordings from the AD group corresponded to an average age of 72 years (57-90 y). The corresponding value for the MCI group was 75 years (53-93 y), and for the NC group, it was 69 years (54-88 y).”

2. Address data imbalance (e.g., female-majority dataset) and its effect on generalization.

Response: This is a vital methodological point. We initially prioritized the female cohort solely to prevent extreme class imbalance, given the severe lack of available male recordings across target cognitive stages in the AI Hub repository. We entirely concur that relying on a female-majority dataset restricts the robust generalization of our trained weights to male demographics, because gender-specific intrinsic vocal characteristics like fundamental frequency and formants impact feature mappings. We have incorporated a critical paragraph inside the Discussion detailing this severe limitation, explicitly proposing subsequent studies incorporating gender-balanced demographic parity. We inserted the following paragraph into the Discussion section just before the "Conclusions" heading:

" Several methodological limitations of this study must be addressed to appropriately contextualize our findings. First, regarding dataset size, while our cross-validation strategy rigorously ensured internal robustness, the overall participant count (N=320) remains relatively limited for training complex transformer architectures like Wav2Vec 2.0. A larger cohort is necessary to fully exploit model capacity and prevent subtle overfitting. Second, we acknowledge a critical gender bias within our sample. We initially prioritized the female cohort solely to prevent extreme class imbalance, given the limited availability of male recordings across target cognitive stages in the AI Hub repository. We acknowledge that relying on a female-only dataset limits the generalizability of our findings to male populations, because gender-specific vocal characteristics such as fundamental frequency and formants significantly affect acoustic feature representations. Consequently, future studies should incorporate gender-balanced cohorts to ensure comprehensive and equitable diagnostic applicability. Finally, the models were trained exclusively on native Korean speakers. Because phonetics, articulation rates, and pausing intervals intrinsically vary across languages, the current findings may not generalize directly to other linguistic populations. Future multinational-scale validations are required.”

3. Describe preprocessing steps such as noise reduction, silence trimming, and normalization.

Response: We thank the reviewer for this important

---

## [Decision Letter · Decision Letter 1]

29 Apr 2026

Prediction of cognitive impairment through speech data analysis: A comparative evaluation of deep learning models

PONE-D-25-44894R1

Dear Dr. Youn,

We’re pleased to inform you that your manuscript has been judged scientifically suitable for publication and will be formally accepted for publication once it meets all outstanding technical requirements.

Kind regards,

Farman Ullah

Academic Editor

PLOS One

Additional Editor Comments (optional):

Reviewers' comments:

Reviewer's Responses to Questions

**Comments to the Author**

1. If the authors have adequately addressed your comments raised in a previous round of review and you feel that this manuscript is now acceptable for publication, you may indicate that here to bypass the “Comments to the Author” section, enter your conflict of interest statement in the “Confidential to Editor” section, and submit your "Accept" recommendation.

Reviewer #1: All comments have been addressed

Reviewer #2: All comments have been addressed

2. Is the manuscript technically sound, and do the data support the conclusions?

Reviewer #1: Yes

Reviewer #2: Yes

3. Has the statistical analysis been performed appropriately and rigorously? 

Reviewer #1: Yes

Reviewer #2: Yes

4. Have the authors made all data underlying the findings in their manuscript fully available?

Reviewer #1: Yes

Reviewer #2: Yes

5. Is the manuscript presented in an intelligible fashion and written in standard English?

Reviewer #1: Yes

Reviewer #2: Yes

6. Review Comments to the Author

Reviewer #1: The authors have undertaken the substantial revision and the manuscript has improved considerably from its original version.

Reviewer #2: (No Response)

7. PLOS authors have the option to publish the peer review history of their article (what does this mean?). If published, this will include your full peer review and any attached files.

Reviewer #1: No

Reviewer #2: No

---

## [Editor Report · Acceptance letter]

PONE-D-25-44894R1

PLOS One

Dear Dr. Youn,

I'm pleased to inform you that your manuscript has been deemed suitable for publication in PLOS One. Congratulations! Your manuscript is now being handed over to our production team.

Kind regards,

on behalf of

Dr. Farman Ullah

Academic Editor

PLOS One